# Exsolution trends and co-segregation aspects of self-grown catalyst nanoparticles in perovskites

Ohhun Kwon[1,*], Sivaprakash Sengodan[1,*], Kyeounghak Kim[2], Gihyeon Kim[1], Hu Young Jeong[3], Jeeyoung Shin[4], Young-Wan Ju[5], Jeong Woo Han[2] & Guntae Kim[1]

In perovskites, exsolution of transition metals has been proposed as a smart catalyst design for energy applications. Although there exist transition metals with superior catalytic activity, they are limited by their ability to exsolve under a reducing environment. When a doping element is present in the perovskite, it is often observed that the surface segregation of the doping element is changed by oxygen vacancies. However, the mechanism of co-segregation of doping element with oxygen vacancies is still an open question. Here we report trends in the exsolution of transition metal (Mn, Co, Ni and Fe) on the $PrBaMn_2O_{5+\delta}$ layered perovskite oxide related to the co-segregation energy. Transmission electron microscopic observations show that easily reducible cations (Mn, Co and Ni) are exsolved from the perovskite depending on the transition metal-perovskite reducibility. In addition, using density functional calculations we reveal that co-segregation of B-site dopant and oxygen vacancies plays a central role in the exsolution.

[1] Department of Energy Engineering, Ulsan National Institute of Science and Technology (UNIST), Ulsan 44919, Republic of Korea. [2] Department of Chemical Engineering, University of Seoul, Seoul 02504, Republic of Korea. [3] UNIST Central Research Facilities and School of Materials Science and Engineering, UNIST, Ulsan 44919, Republic of Korea. [4] Division of Mechanical Systems Engineering, Sookmyung Women's University, Seoul 04310, Republic of Korea. [5] Department of Chemical Engineering, Wonkwang University, Iksan 54538, Republic of Korea. * These authors contributed equally to this work. Correspondence and requests for materials should be addressed to J.W.H. (email: jwhan@uos.ac.kr) or G.K. (email: gtkim@unist.ac.kr).

Perovskites, a class of metal oxides with well-defined structures, have recently occupied a predominant position within the portfolio of compounds that have been explored as the electrode materials for fuel cells, electronic devices, heterogeneous catalysis in syngas production and components for solar cells[1–4]. This wide variety of properties originates from the exceptional structural and compositional flexibility of the perovskite structures. In recent years, composite materials realized by integration of functional catalyst nanoparticles with perovskite oxide supports have received rising attention. The nanoparticle-supported perovskite oxide materials can be prepared by conventional deposition methods, such as wet impregnation or vapour deposition[5–7]. Although these techniques are applied widely, controllable anchoring still encounters many challenges. For example, a wet impregnation technique always suffers from coarsening and agglomeration of the catalyst nanoparticles on the surface of the perovskite, leading to severe cell degradation. Therefore, an advanced approach to prepare well-defined nanoparticle-supported perovskites is required to overcome the drawbacks of conventional methods.

Exsolution based on in situ growth of metal nanoparticles from the parent perovskite is an attractive approach for designing nanoparticle-supported perovskite materials. The catalytically active transition metals, such as Pd, Ru, Pt, Co and Ni, are incorporated on the B site of perovskite oxide ($ABO_3$) during material synthesis in air, and then the transition metals are exsolved from the perovskite backbone as highly dispersed nanoparticles under a reducing atmosphere[8–11]. The exsolved nanoparticles are socketed on the surface of the perovskite, preventing agglomeration and coarsening of the nanoparticles during operation conditions[12]. Furthermore, the Irvine group used a different strategy through the control of non-stoichiometry (A-site deficiency in the $ABO_3$ stoichiometry) to promote exsolution[13]. Well-defined nanoparticle-supported perovskites have been obtained in these reports; however, the exsolution trends of transition metals is still scarce and has been focussed on simple perovskites.

Recently, layered perovskite structures have received considerable attention because of their interesting properties, such as high electrical conductivity, fast surface oxygen exchange and easy oxygen ion diffusion[14–17]. However, there have been very few reports focussed on exsolution in layered perovskites due to the lack of redox stable layered perovskites. Therefore, exsolution trends in redox stable layered perovskites are of particular interest, not only from scientific but also from engineering points of view, because they can provide a strategy of tailoring materials for fuel cell electrodes, catalytic oxidation of hydrocarbon and thermochemical hydrogen production from water[18,19].

Here we report the contribution of various transition metals for in situ growth of finely dispersed metal nanoparticles on a $PrBaMn_{1.7}T_{0.3}O_{5+\delta}$ (T = Mn, Co, Ni, and Fe) layered perovskite with the aim to verify trends in exsolution and improve the electrochemical performance of solid oxide fuel cell anodes. The exsolution trends of the B-site dopants (Mn, Co, Ni and Fe) are verified by a transmission electron microscopy (TEM) analysis and density functional theory (DFT) calculations.

## Results

**Structure and morphological characterization**. The crystalline structures of the oxide materials before and after reduction were examined using the X-ray diffraction technique. As shown in Supplementary Fig. 1, diffraction patterns for all samples sintered at 950 °C in air exhibit a simple perovskite structure with a mixture of cubic and hexagonal phases without any secondary phases[1]. Apparently, the B-site doping has no influence on the formation of the simple perovskite structure. Supplementary Figure 2 shows the X-ray diffraction patterns of $PrBaMn_2O_{5+\delta}$ (L-PBMO), $PrBaMn_{1.7}Co_{0.3}O_{5+\delta}$ (L-PBMCO), $PrBaMn_{1.7}Ni_{0.3}O_{5+\delta}$ (L-PBMNO) and $PrBaMn_{1.7}Fe_{0.3}O_{5+\delta}$ (L-PBMFO) after reduction in humidified $H_2$ (3% $H_2O$) at 800 °C for 4 h. The reduced samples present a single phase of the layered perovskite structure with metal or metal oxide phases, indicating that the phase transition from the simple perovskite to the layered perovskite and exsolution occurred in the reducing atmosphere. Although all the samples were reduced under the same conditions, MnO, metallic Co and Ni phases are observed in the L-PBMO, L-PBMCO and L-PBMNO, respectively, and no Fe phase is observed in the L-PBMFO. These results clearly show that MnO, Co and Ni are more easily exsolved to form nanoparticles than Fe, and thus B-site transition metals show some trend to exsolve in the layered perovskite oxide.

Supplementary Figure 3 shows scanning electron microscope (SEM) images of (a) L-PBMO, (b) L-PBMCO, (c) L-PBMNO and (d) L-PBMFO after reducing treatment using humidified (3% $H_2O$) $H_2$ at 800 °C. As shown in the SEM images, the surface morphologies of the reduced samples are similar, and some small spherical nanoparticles with 20–50 nm diameter are only observed on the surface of L-PBMCO and L-PBMNO. Although Co and Ni nanoparticles are readily observed in the SEM images, this is not straightforward in the case of MnO because the population of the exsolved MnO nanoparticles is small in dimension and low in number.

To observe the exsolved nanoparticles and morphologies of the reduced samples in detail, we measured TEM. Energy dispersive spectroscopy (EDS) revealed that Mn and O elements coexist in the L-PBMO (Supplementary Fig. 4), speculating that MnO nanoparticles are exsolved in reducing atmosphere as seen in the X-ray diffraction results. In the bulk state, MnO exsolution is not easily detectable because both MnO and L-PBMO are oxide materials[13]. To realize the nature of exsolved MnO nanoparticles, a $Pr_{0.5}Ba_{0.5}MnO_3$ polycrystalline film was fabricated by pulse laser deposition (PLD) followed by reduction in $H_2$ at 800 °C for the exsolution of the nanoparticles from the lattice. From the bright-field TEM image (Supplementary Fig. 5a), it is observed that the nanoparticles having roughly 40 nm diameter are successfully exsolved from the L-PBMO in the reducing atmosphere. For the exsolved nanoparticles of L-PBMO, the lattice space between planes is identified as 0.225 nm by high-resolution TEM and Fast Fourier Transformation (Supplementary Fig. 5b), corresponding to the lattice constant of (200) planes of the MnO, which is in agreement with the X-ray diffraction results (Supplementary Fig. 2a). To identify the oxidation state of Mn nanoparticles, electron energy-loss spectroscopy was performed and the results indicated that O-K and $Mn-L_{2,3}$ features are consistent with MnO, as reported in a previous study (Supplementary Fig. 5c)[20,21]. Based on the relative reducibility and thermodynamic possibility, Mn exsolution from the L-PBMO would be MnO rather than metallic Mn[22]. Generally, among manganese oxides there are four relevant oxides (MnO, $Mn_3O_4$, $Mn_2O_3$ and $MnO_2$) in the redox cycling reactions. Among them, $Mn_3O_4$ (equation (1)) and $Mn_2O_3$ (equation (2)) are easily reduced to MnO in a reducing atmosphere, and the reaction is identified by the values of Gibbs energy $\Delta G_r$. In the case of MnO, however, the reduction of MnO to metallic Mn (equation (3)) is thermodynamically unfavourable at 1,000 K due to the positive value of $\Delta G_r$ for MnO reduction.

$$Mn_3O_4(s) \underset{1,000K, \ H_2}{\longleftrightarrow} 3MnO(s) + H_2O(g) \ \Delta G_r = -86.63 \ kJ \ mol^{-1}$$

$$(1)$$

$$Mn_2O_3(s) \underset{1,000K, \; H_2}{\longleftrightarrow} 2MnO(s) + H_2O(g) \quad \Delta G_r < -86.63 \, kJ \, mol^{-1}$$
$$(2)$$

$$MnO(s) \underset{1,000K, \; H_2}{\longleftrightarrow} Mn(s) + H_2O(g) \quad \Delta G_r > 100 \, kJ \, mol^{-1} \quad (3)$$

Figure 1 shows bright-field TEM images and high-resolution TEM images of reduced L-PBMCO and L-PBMNO. As shown in Fig. 1a,d, the morphologies of L-PBMCO and L-PBMNO are similar to SEM images. From the high-resolution TEM images (Fig. 1b,e), it is observed that the nanoparticles having roughly 30 nm diameter are successfully exsolved from the L-PBMCO and L-PBMNO, respectively, in reducing atmosphere. In addition, the lattice spaces between planes of exsolved nanoparticles are 0.204 nm (Fig. 1c) and 0.176 nm (Fig. 1f), and these values are consistent with the lattice constant of $(11\bar{1})$ planes of Co metal and (200) planes of Ni metal, respectively. As shown in Fig. 1g,h, the EDS micrograph and elemental mapping also reveal that Co and Ni nanoparticles are exsolved from L-PBMCO and L-PBMNO, respectively. Unlike L-PBMO, metal nanoparticles are exsolved without MnO from L-PBMCO and L-PBMNO, which is also supported by thermodynamic possibilities. The reductions of CoO and NiO to metallic Co and Ni are thermodynamically favourable at 1000 K due to the negative value of $\Delta G_r$ for the CoO (equation (4)) and NiO (equation (5)) reductions[23]. As noted from the X-ray diffraction results, MnO

nanoparticles are exsolved from L-PBMFO in humidified $H_2$ (3% $H_2O$) at 800 °C, which is also confirmed by the TEM–EDS analysis results (Supplementary Fig. 6).

$$CoO(s) \underset{1,000K, \; H_2}{\longleftrightarrow} Co(s) + H_2O(g) \quad \Delta G_r = -29.172 \, kJ \, mol^{-1}$$
$$(4)$$

$$NiO(s) \underset{1,000K, \; H_2}{\longleftrightarrow} Ni(s) + H_2O(g) \quad \Delta G_r = -43.48 \, kJ \, mol^{-1} \quad (5)$$

X-ray photoelectron spectroscopy (XPS) was performed to examine oxidation states of B-site dopants in the L-PBMO, L-PBMCO, L-PBMNO and L-PBMFO. As shown in Supplementary Fig. 7a, the binding energy peaks of metallic Ni (855.3 and 873.1 eV) could be detected in L-PBMNO[24], which is consistent with the X-ray diffraction and TEM results. Furthermore, the percentages of metallic Ni and $Ni^{2+}$ are about 58% and 42%, respectively, indicating that approximately 58 % of Ni migrates to the surface in L-PBMNO. Supplementary Figure 7b shows the binding energy peaks of $Mn^{2+}$ (641 and 652.8 eV) and $Mn^{3+}$ (642.8 and 653.7 eV) in L-PBMO[25]. Supplementary Figure 7c shows two major peaks with binding energy at 710.2 and 723.9 eV, corresponding to Fe $2p_{3/2}$ and Fe $2p_{1/2}$, accompanied by two shake-up satellite peaks (718.6 and 732 eV) in L-PBMFO. The appearance of two peaks at 710 and 723.7 eV is $Fe^{2+}$, whereas the other two peaks at 712.5 and

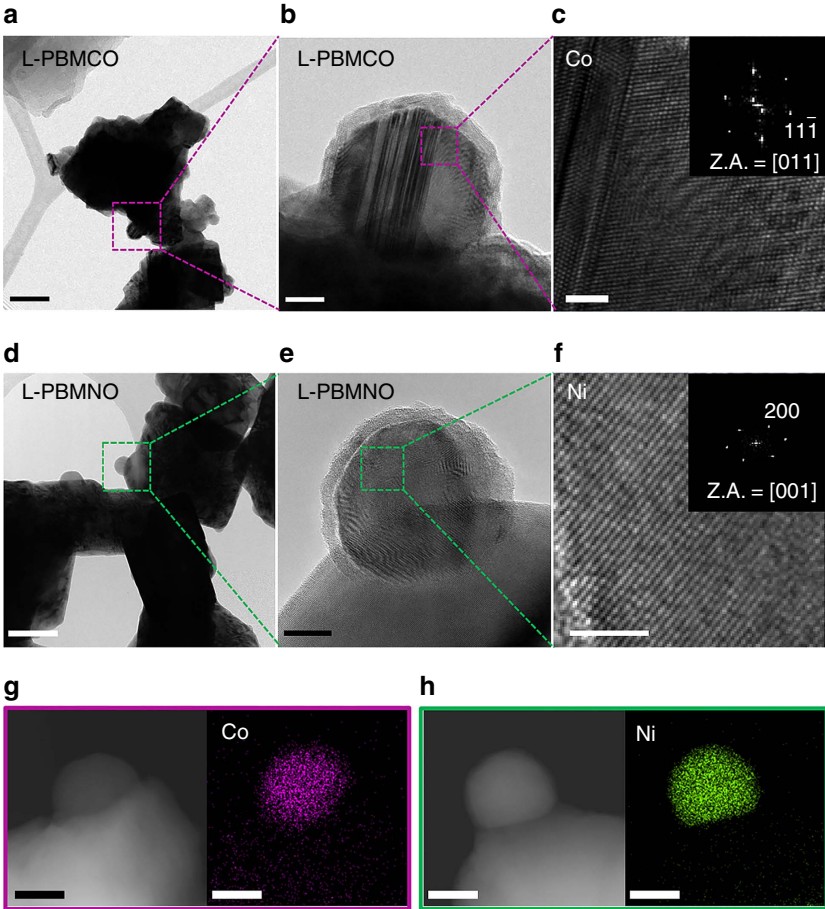

**Figure 1 | Transmission electron microscopic analysis.** (**a**) A bright-field (BF) TEM image; scale bar, 100 nm and (**b**) high-resolution (HR) TEM image of $PrBaMn_{1.7}Co_{0.3}O_{5+\delta}$ (L-PBMCO) sample; scale bar 10 nm. (**c**) Magnified HR TEM image of exsolved Co nanoparticle; scale bar 2 nm, (**d**) BF TEM image; scale bar 100 nm and (**e**) HR TEM image of $PrBaMn_{1.7}Ni_{0.3}O_{5+\delta}$ (L-PBMNO) sample; scale bar 10 nm. (**f**) Magnified HR TEM image of Ni nanoparticle; scale bar 2 nm. (**g**) High-angle annular dark-field (HAADF) image of the L-PBMCO with the EDS elemental map of Co; scale bar 25 nm. (**h**) HAADF image of the L-PBMNO with the EDS elemental map of Ni; scale bar 25 nm.

725.5 eV are characteristic of $Fe^{3+}$. However, in the L-PBMCO, it is not easy to identify exact oxidation state of Co because binding energy peaks of Co 2p and Ba 3d main lines overlap each other (Supplementary Fig. 7d). From the XPS results, we can identify that Mn and Fe have no metallic phase in L-PBMO and L-PBMFO, respectively, which is also in agreement with our experimental results.

**The effect of co-segregation energy on exsolution.** A question that remains to be addressed is how the exsolution phenomenon occurs in layered perovskites. The exsolution phenomenon was confirmed in A-site-deficient simple perovskites, where oxygen vacancies were introduced during reduction, which destabilizes the perovskite structure and results in spontaneous exsolution of B-site cation. We speculate that spontaneous exsolution phenomenon occurs in layered perovskites when considerable amounts of oxygen vacancies and B-site metal vacancies could be introduced instantaneously (co-segregation) by reduction, and then the metal oxide can be converted to the corresponding metal or metal oxide (Fig. 2). This mechanism of exsolution in layered perovskites is expressed as point defect (Schottky-type defect) reactions as follows:

$$T_{Mn}^{\times} + O_O^{\times} \leftrightarrow TO + V_O^{\circ\circ} + V_{Mn}^{''} \tag{6}$$

$$TO \leftrightarrow T_{(metallic\ exsolution)} + \frac{1}{2}O_2 \tag{7}$$

where $T_{Mn}^{\times}$ denotes the B-site dopant in the Mn site with the net charge zero, $O_O^{\times}$ denotes oxygen in the oxygen site with the net charge zero, $V_O^{\circ\circ}$ denotes the oxygen ion vacancy with the net charge $+2$, $V_{Mn}^{''}$ denotes the cation vacancy in the Mn site with the net charge $-2$, and TO denotes the transition metal oxide.

Based on the X-ray diffraction and TEM results, it appears that transition metals show different degrees of exsolution in the B site of the layered perovskite. To verify the exsolution trends of B-site transition metals in the layered perovskite, we performed DFT calculations. As mentioned above, it can be thought that the exsolution process occurs through two key sequential steps: (1) metal segregation towards the surface and (2) phase transition from the segregated phase to metallic phase.

To quantitatively compare the exsolution between the Mn (non-doped L-PBMO) and other B-site dopants (L-PBMCO, L-PBMNO, L-PBMFO), we calculated the co-segregation energy of B-site transition metal accompanying oxygen vacancies. This approach involves the assumption that the oxygen vacancy co-segregates with the nearby B-site metal towards the surface (Fig. 3a). In a previous study, Hamada et al.[26] performed DFT calculations to identify the role of the oxygen vacancy on precious metal (Pd, Pt and Rh) exsolution. The introduction of oxygen vacancies significantly enhanced the Pd exsolution, stabilizing the surfaces over a wide range of oxygen chemical potentials on $LaFe_xPd_{1-x}O_3$. Neagu et al.[13] also reported that non-stoichiometry such as A-site deficiency facilitates the formation of oxygen vacancies, which results in particle exsolution on $La_{\alpha}Sr_{1-\alpha}Ti_{\beta}Ni_{1-\beta}O_{3-\gamma}$ surfaces. These studies imply that the creation of oxygen vacancies at the surfaces or bulk of the perovskite oxides is closely related to the exsolution of metal or metal oxide from the parent lattice. Since these considerations contain both the effect of B-site metal segregation and vacancy formation, it enables us to make a comprehensive comparison of the tendency of B-site transition metal exsolution. The co-segregation energies obtained by DFT calculations are $-0.47$, $-0.55$, $-0.50$ and $-0.15\ eV$ for L-PBMO, L-PBMCO, L-PBMNO and L-PBMFO, respectively (Fig. 3b), indicating that Co and Ni more favourably exsolve towards the surface than Mn and Fe less favourably exsolves than Mn.

Once the B-site metal and oxygen vacancy co-segregate towards the surface, stabilization of the segregated phase containing the B-site dopant on the surface is required to maintain the B-site dopant as metallic phase. For this, there should be a preference of oxygen vacancy formation near the B-site dopant compared to other sites. Therefore, we examined the oxygen vacancy formation at various lattice O sites in Supplementary Fig. 8. Our results showed that the most stable sites of oxygen vacancy formation in L-PBMO are near the surface (Supplementary Table 1). Thus the oxygen vacancy formed in the bulk prefers to be segregated out to the surface. Then we compared the oxygen vacancy-formation energy of B-site dopants at the surface of layered perovskites (Supplementary Fig. 9). The oxygen vacancy-formation energies

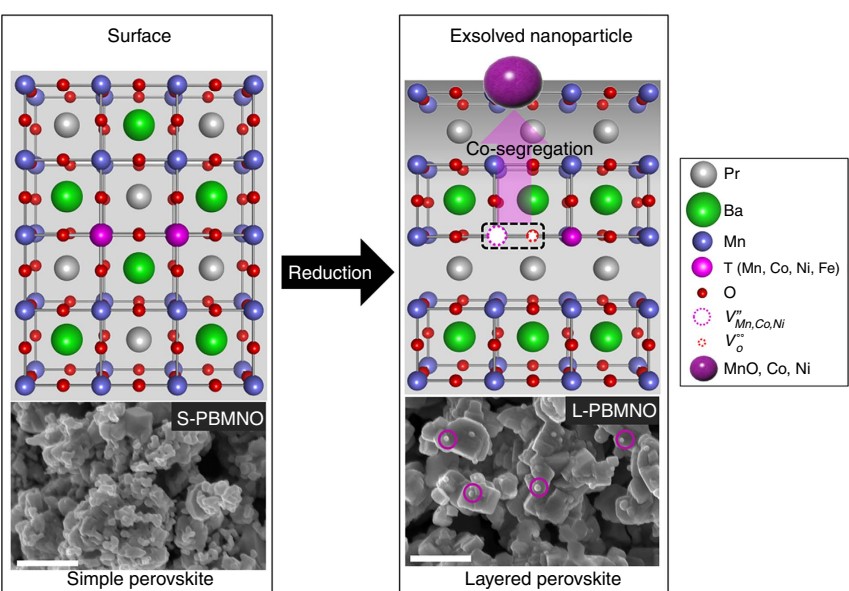

**Figure 2 | Exsolution of B-site cation with oxygen from layered perovskite in a reducing atmosphere.** The SEM images present surface morphologies of $Pr_{0.5}Ba_{0.5}Mn_{0.85}Ni_{0.15}O_3$ before reduction and $PrBaMn_{1.7}Ni_{0.3}O_{5+\delta}$ after reduction in humidified (3% $H_2O$) $H_2$ at 800 °C for 4 h; scale bar 500 nm. In the SEM image of $PrBaMn_{1.7}Ni_{0.3}O_{5+\delta}$, the purple circles indicate the exsolved nanoparticles.

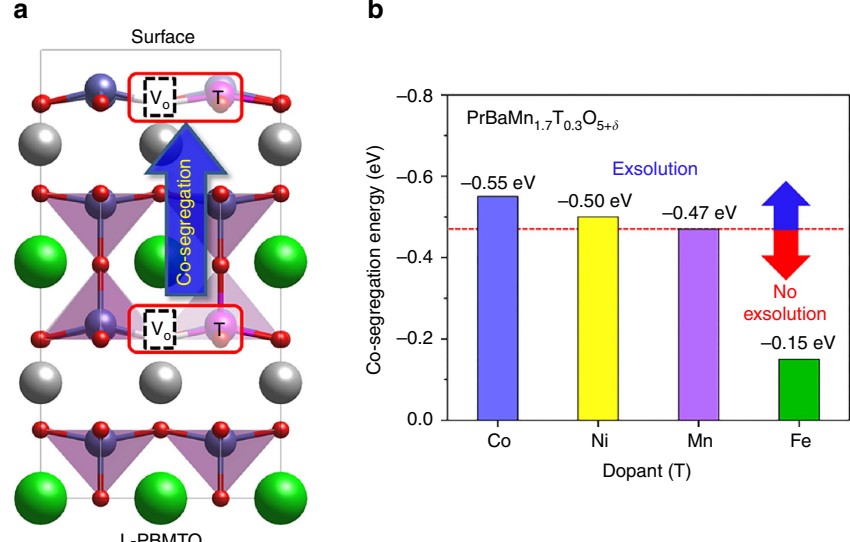

**Figure 3 | Density functional theory calculations for elucidating co-segregation energy.** (**a**) Schematic illustration of our model used for the calculations of co-segregation energy. Pr, Ba, Mn, T (Mn, Co, Ni and Fe) and O atoms are shown as grey, green, dark blue, purple and red, respectively. The inset red boxes indicate the co-segregation of B-cation with an oxygen vacancy. (**b**) Comparison of the co-segregation energy with the dopant (T) materials.

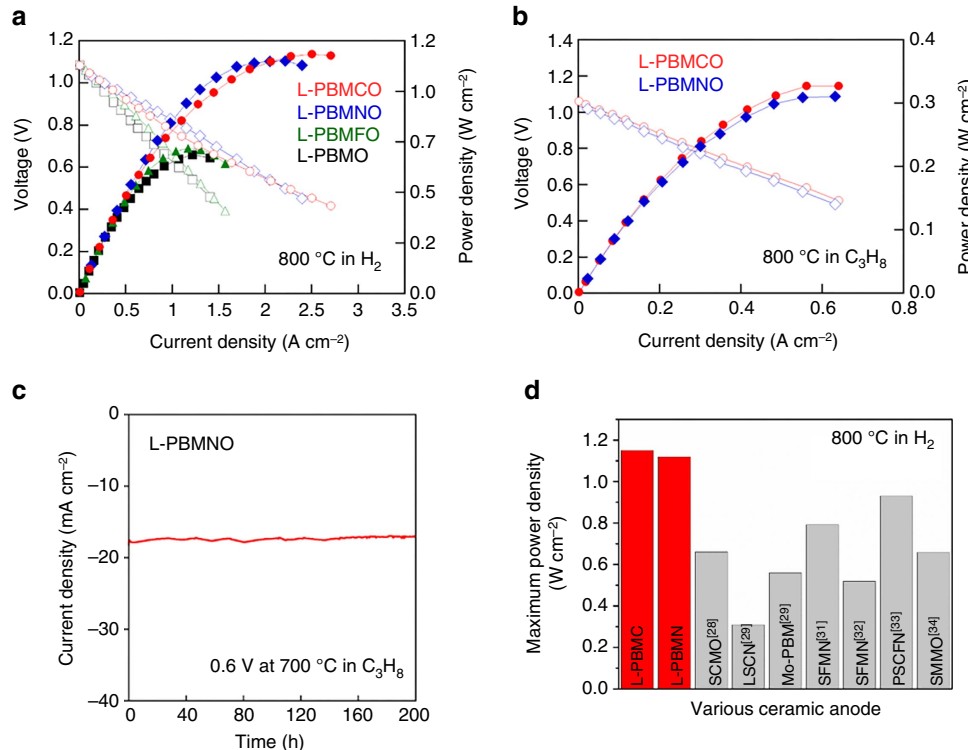

**Figure 4 | Electrochemical properties of layered PrBaMn$_{1.7}$T$_{0.3}$O$_{5+\delta}$ anode in fuel cells.** (**a**) I–V curve and the corresponding power densities of the L-PBMTO (T = Mn, Co, Ni and Fe) electrode using humidified (3% H$_2$O) H$_2$ and ambient air as the oxidant at 800 °C. (**b**) I–V curve and the corresponding power densities of the PrBaMn$_{1.7}$Co$_{0.3}$O$_{5+\delta}$ (L-PBMCO) and PrBaMn$_{1.7}$Ni$_{0.3}$O$_{5+\delta}$ (L-PBMNO) electrode using C$_3$H$_8$ as fuel and ambient air as the oxidant at 800 °C. (**c**) Electrochemical performances of L-PBMNO anode in C$_3$H$_8$ at 700 °C under a constant voltage of 0.6 V. (**d**) Comparison of the maximum power density at 800 °C in H$_2$ of the present work and other studies in the literature[28–34].

of the L-PBMCO (2.46 eV), L-PBMNO (2.85 eV) and L-PBMFO (2.91 eV) surfaces are lower than the L-PBMO surface (2.97 eV). This indicates that Co, Ni and Fe create oxygen vacancies more easily than Mn at the surface, and consequently the segregated phase is also easily stabilized to the metallic phase in the following order: L-PBMCO > L-PBMNO > L-PBMFO > L-PBMO. However, L-PBMFO has lower co-segregation energy than

L-PBMO as confirmed previously, it is expected that no exsolved metallic Fe nanoparticles exist on the surface of L-PBMFO, which is in good agreement with our experimental results.

**Power output and durability of fuel cells.** To confirm the catalytic effect of the exsolved nanoparticles, the electrochemical performance of single cells was tested using

$La_{0.9}Sr_{0.1}Ga_{0.8}Mg_{0.2}O_{3-\delta}$ (LSGM) electrolyte-supported cell in humidified $H_2$ (3% $H_2O$) as the fuel and ambient air as the oxidant with the configuration of L-PBMTO (T = Mn, Co, Ni, and Fe)/$La_{0.4}Ce_{0.6}O_{2-\delta}$ (LDC)/LSGM/$NdBa_{0.5}Sr_{0.5}Co_{1.5}Fe_{0.5}O_{5+\delta}$ -$Ce_{0.9}Gd_{0.1}O_{2-\delta}$. The electrochemical impedance spectra of single cell corresponding to the I–V polarization curve at 800 °C are presented in Supplementary Fig. 10 and Fig. 4a. The non-ohmic resistances of L-PBMO, L-PBMCO, L-PBMNO and L-PBMFO are 0.265, 0.167, 0.099 and 0.221 $\Omega\, cm^2$ at 800 °C, respectively. In previous reports, exsolved Co and Ni nanoparticles increase the catalytic activation in fuel oxidation, which reduces the anode polarization resistance[9,27]. Therefore, the non-ohmic resistances of L-PBMCO and L-PBMNO are lower than L-PBMO and L-PBMFO. The maximum power densities of the L-PBMO, L-PBMCO, L-PBMNO and L-PBMFO single cells are 0.661, 1.15, 1.12 and 0.690 $W\, cm^{-2}$ at 800 °C in $H_2$, respectively. Figure 4b shows the I–V polarization curves of the L-PBMCO (0.331 $W\, cm^{-2}$) and L-PBMNO (0.322 $W\, cm^{-2}$) single cells in $C_3H_8$ at 800 °C. Supplementary Table 2 shows the comparison of electrochemical performance of other ceramic anodes decorated with metal particles in hydrocarbon fuels. Furthermore, no remarkable degradation was observed under a constant voltage of 0.6 V at 700 °C in $C_3H_8$ (Fig. 4c) and $H_2$ (Supplementary Fig. 11) for > 200 h. It is worthwhile to point out that even though oxidation catalysts, such as Ce, Pt and metal alloys, were not added into either of the electrodes from the outside, the exsolved nanoparticles on the surface of the ceramic anodes serve as a good fuel oxidation catalyst. Especially, the exsolved Co and Ni nanoparticles on the surface of the layered perovskite show excellent cell performance (1.15 and 1.12 $W\, cm^{-2}$ at 800 °C in $H_2$, respectively) among developed ceramic anodes without adding any catalysts externally (Fig. 4d and Supplementary Table 3)[28–34].

## Discussion

In summary, the present work demonstrates the transition metal (Mn, Co, Ni and Fe) exsolution trends for self-grown catalytic nanoparticle on a layered perovskite, which may be useful for the development of tailored functional materials. On the basis of DFT calculations, we proposed a possible mechanism for the exsolution of transition metals in layered perovskites, wherein co-segregation of B-site metal and oxygen vacancies plays a central role in the exsolution. We found that Co and Ni have high co-segregation energy ($-0.55$ and $-0.50$ eV) in the layered perovskite, which facilitates the exsolution of Co and Ni metal particles on the surface. The maximum power densities of an electrolyte-supported cell with L-PBMCO and L-PBMNO anodes reached 1.15 and 1.12 $W\, cm^{-2}$ in humidified $H_2$ at 800 °C, respectively, constituting excellent electrochemical performance as compared to other ceramic anodes. Our findings thus provide a key to understand the exsolution trends in transition metals (Mn, Co, Ni and Fe) containing perovskites and design highly catalytic perovskite oxides for fuel reforming and electro-oxidation.

## Methods

**Material synthesis.** $Pr_{0.5}Ba_{0.5}Mn_{0.85}T_{0.15}O_{3-\delta}$ (T = Mn, Co, Ni and Fe) powders were prepared by the Pechini method. Stoichiometric amounts of $Pr(NO_3)_3 \cdot 6H_2O$ (Aldrich, 99.9%, metal basis), $Ba(NO_3)_2$ (Aldrich, 99 + %), $Mn(NO_3)_2 \cdot 4H_2O$ (Aldrich, 98%), $Co(NO_3)_2 \cdot 6H_2O$ (Aldrich, 98 + %), $Ni(NO_3)_2 \cdot 6H_2O$ (Aldrich, 98.5 + %) and $Fe(NO_3)_3 \cdot 9H_2O$ (Aldrich, 98 + %) were dissolved in distilled water with proper amounts of ethylene glycol and citric acid, followed by combustion to obtain fine powders. These powders were calcined at 600 °C for 4 h and then sintered at 950 °C for 4 h in air. The A-site-layered $PrBaMn_{1.7}T_{0.3}O_{5+\delta}$ (T = Mn, Co, Ni, and Fe) was obtained by annealing $Pr_{0.5}Ba_{0.5}Mn_{0.85}T_{0.15}O_{3-\delta}$ (T = Mn, Co, Ni and Fe) oxide, respectively, at 800 °C for 4 h in humidified $H_2$. The chemical composition of the synthesized powders and their abbreviations are given in

Supplementary Table 4. A thin $Pr_{0.5}Ba_{0.5}MnO_3$ film was deposited on the dense $Al_2O_3$ substrate by PLD by using a commercial system (PLD-7; PASCAL, Japan). The oxygen pressure was adjusted to 0.67 Pa before the deposition process by introducing commercially available oxygen (without further purification), and the substrate was heated to 800 °C by using an infrared heater. An excimer laser was used with a power of 180 mJ pulse$^{-1}$ and a frequency of 10 Hz to deposit the film.

**Structure characterization.** The crystal structures of the samples were identified by X-ray powder diffraction (Rigaku-diffractometer, Cu Kα radiation, 40 kV, 30 mA). The morphologies of the anode materials were investigated using a field emission SEM. TEM images were obtained with a JEOL JEM 2100F with a probe forming (STEM) Cs (spherical aberration) corrector at 200 kV. Cross-sectional samples for the TEM analysis were prepared by using a focussed ion beam (Helios 450HP, FEI). XPS analyses were conducted on ESCALAB 250XI from Thermo Fisher Scientific with a monochromated Al-Kα (ultraviolet He1, He2) X-ray source. The base pressure inside the spectrometer during analysis was $1 \times 10^{-10}$ mm Hg.

**Computational methods.** DFT calculations were carried out using the Vienna *ab initio* Simulation Package[35]. Exchange-correlation energies were treated by the Perdew–Burke–Ernzerhof functional based on generalized gradient approximation (GGA). A plane wave expansion with a cutoff of 400 eV was used with a $3 \times 3 \times 1$ Monkhorst–Pack k-point sampling of the Brillouin zone for all slab model calculations[36]. Gaussian smearing was used with a width of 0.05 eV to determine partial occupancies. Geometries were relaxed using a conjugate gradient algorithm until the forces on all unconstrained atoms were $< 0.03$ eV Å$^{-1}$. In order to take into account on-site Coulomb and exchange interactions, we used GGA + U schemes with the effective U values of 4.0, 3.3, 6.4 and 4.0 to Mn, Co, Ni and Fe, respectively[37]. Based on our X-ray diffraction and TEM images, the L-PBMO structure was optimized with a tetragonal $P4/mmm$ structure ($a = b = 4.035$, $c = 7.940$) (Supplementary Fig. 12a). An eight-layered PBMO slab model was also constructed with vacuum thickness of up to 17 Å, which was sufficient to describe surface phenomena (Supplementary Fig. 12b).

The oxygen vacancy-formation energy ($E_{vf}$) was calculated from the total energies of the supercells with various defect positions (Supplementary Fig. 8).

$$E_{vf} = (E_{vac} + \frac{1}{2}E_{O_2}) - E_{clean}, \tag{1}$$

where $E_{vac}$ is the total energy of the system containing an oxygen vacancy, $E_{O_2}$ is the total energy of an isolated oxygen molecule in the gas phase, and $E_{clean}$ is the total energy of optimized perfect slab structures. The segregation energy of B-site metal ($E_{seg}$) with or without oxygen vacancies is defined as the total energy difference between the systems with the exsoluting B-site metal located at the surface and in the bulk:

$$E_{seg} = E_{(B-Vo)\_surf} - E_{(B-Vo)\_bulk}, \tag{2}$$

where $E_{(B-Vo)\_surf}$ and $E_{(B-Vo)\_bulk}$ are the total energy of the exsoluting B-site metal located at the surface and in the bulk with or without oxygen vacancies, respectively. With our definition, a negative segregation energy indicates that the B metal energetically prefers to exsolve towards the surface.

**Fabrication of fuel cells.** LSGM powder was prepared by conventional solid state reaction and electrolyte substrate was prepared by pressing and followed by sintering at 1,475 °C. Stoichiometric amounts of $La_2O_3$ (Sigma 99.99%), $SrCO_3$ (Sigma, 99.99%), $Ga_2O_3$ (Sigma, 99.99%) and MgO (Sigma, 99.9%) powders were ball milled in ethanol for 24 h. After drying, the powder was calcined at 1,000 °C for 6 h. The thickness of LSGM electrolyte was polished about 250 μm. LDC was also prepared by ball milling stoichiometric amounts of $La_2O_3$ and $CeO_2$ (Sigma, 99.99%) in ethanol and then calcined at 1,000 °C for 6 h. For preparation of the anode ink, $Pr_{0.5}Ba_{0.5}Mn_{0.85}T_{0.15}O_3$ (T = Mn, Co, Ni and Fe) was mixed with an organic binder (Heraeus V006) (1:2 weight ratio). $NdBa_{0.5}Sr_{0.5}Co_{1.5}Fe_{0.5}O_{5+\delta}$-$Ce_{0.9}Gd_{0.1}O_{2-\delta}$ cathode ink was prepared by precalcined cathode and GDC powders (at a weight ratio of 60:40) were mixed using ball milling, together with an organic binder. The electrode inks were applied on the LSGM pellet by screen printing method and then calcined at 950 °C in air for 4 h. The porous electrode had an active area of 0.36 $cm^2$ and thickness about 20 μm. LDC layer was used as the buffer layer between the anode and the electrolyte to prevent interdiffusion of ionic species between anode and LSGM electrolyte. For fuel cell performance tests, the cells were mounted on alumina tubes with ceramic adhesives (Ceramabond 552, Aremco). Silver paste and silver wire were used for electrical connections to both the anode and the cathode. The entire cell was placed inside a furnace and heated to the desired temperature. I–V polarization curves were measured using a BioLogic Potentiostat.

**Data availability.** The data that support the findings of this study are available from the corresponding authors upon reasonable request.

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

## Acknowledgements

This research was supported by the New & Renewable Energy Core Technology Program of the Korea Institute of Energy Technology Evaluation and Planning (KETEP), granted financial resource from the Ministry of Trade, Industry & Energy, Republic of Korea (No. 20143030031430), the Mid-career Researcher Program (NRF-2015R1A2A1A10055886) and the Korea Research Fellowship Program (2016H1D3A1909709) through the National Research Foundation of Korea, funded by the Ministry of Science, ICT and Future Planning. This research was supported by Basic Science Research Program through the National Research Foundation of Korea (NRF) funded by the Ministry of Education (2016-0790). This study was supported by Samsung Research Funding Center for Future Technology (G01150336).

## Author contributions

O.K., S.S. and G.K. contributed to fabricate sample and performed the electrochemical experiments. Y.-W.J. fabricated PLD thin film samples. H.Y.J. and O.K. collected and analysed the TEM. K.K. performed the DFT calculations. All authors contributed to writing the paper. J.S., Y.-W.J., J.W.H. and G.K. conceived and designed the experiment and analysed the data.

## Additional information

**Competing interests:** The authors declare no competing financial interests.

