## [Peer Review File · Nature Communications]

Reviewers' comments:

Reviewer #1 (Remarks to the Author):

This manuscript discusses the "periodicity" in the exsolution of transition metal nanoparticle on the layered perovskite oxide and the co-segregation energy. While the first concept is elusive, the second one has been proposed by others. While I believe that the study provides significant new insight on the exsolution process, there are some critical issues need to be considered. I list them here:

1. At line 77, the authors claim there isn't any secondary phase, however the hexagonal phase is indeed a secondary phase.
2. At line 88, it is easy to understand that MnO, Co, and Ni are more easily exsolved than Fe. However, it is not clear what the authors mean by "periodicity" as described in the following sentence.
3. At line 93 and Figure 2, the SEM image after exsolution shows not only the exsolution of the nanoparticles but also a flatter surface. The small clusters of the host material also vanish after exsolution. This is not in agreement with the authors' comment that the surfaces are similar. Explanations should be provided.
4. At line 152, the authors assume that the exsolution has two steps, where the metal oxide is formed first. Although this is hard to probe using DFT calculations, I suggest a more careful assumption.
5. At line 178 and Figure 2, the author seems to only consider the segregation on the (001) plane. This is especially doubtful since PBM is a layered double perovskite and the oxygen transportation may differ quite a lot along different orientations. Calculation for other planes should be provided.
6. The more favorable presence of vacancies at the surface may lead to a coulombic repulsion with other positively charged defects and further promote the BaO formation. This should be explained in greater detail as I believe it is important for the clarification of the mechanism.
7. It would be useful to know which percentage of the B-site dopant migrates to the surface as this information could elucidate if there is an over-stoichiometry of Pr and Ba on the surface itself. Also, why don't BaO or Pr₂O₃ form on the surface as a consequence of exsolution? (this could be easily characterized by XPS).

Reviewer #2 (Remarks to the Author):

NCOMMS-16-28049-T

This paper deals with the exsolution of transition metals in layered perovskites. The novel aspect should regard the so called "periodicity of exsolution" in redox stable layered perovskites i.e. a correlation between the thermodynamic reducibility of transition metal cations used as B-site dopants and the occurrence of the corresponding metal and oxide nanoparticles on the surface. My comments are provided below for the authors' consideration.

1. The relationship between the thermodynamic reducibility of transition metals cations used as B-site dopants and the occurrence of the corresponding metal and oxide nanoparticles on the surface is not unexpected. At high temperature, it is obvious that solid state reactions are essentially governed by thermodynamics. However, the authors have carried out a systematic work providing a wide demonstration of what is generally expected and experimentally verified in specific cases in other reports. Anyhow, the general analysis here provided could be of interest for SOFC developers.
2. Whereas the effect of oxygen vacancies and A-site deficiency has been previously investigated (e.g. ref. 12) and the new contribution provided in this regard by this paper, using DFT calculations, should be better elucidated in comparison to the previous literature reports. Moreover, in a previous work recently published, density functional theory calculations have also proved that the crystal reconstruction induces the loss of coordinated oxygen surrounding B-site

cations, serving as the driving force for steering fast nanoparticle growth (Nano Lett., 2016, 16 (8), pp 5303–5309). This previous work is not mentioned in the manuscript.

3. The occurrence of metal or oxide nanoparticle as consequence of the exsolution is essentially demonstrated in this work by using X-ray diffraction and transmission electron microscopy with EDS analysis. TEM analysis provides a local information involving just a few nanoparticles. However, there is no proper analysis of the surface modifications on a statistical basis after the exsolution in terms of composition and oxidation states. On the other hand, a quantitative determination of the surface atomic composition and oxidation states may be provided by X-ray photoelectron spectroscopy analysis. The catalytic activity is essentially related the surface characteristics since reactions occur at the interface; thus, I strongly recommend a surface analysis e.g. by XPS for this work.

4. The aim of using catalysts made of perovskite with exsolved transition metal on the surface in SOFCs is to provide an enhanced activity, being this promoted by the metal nanoparticles on the surface, while keeping good stability towards redox and thermal cycles. The performance of hydrogen-fed SOFCs for the perovskites containing exsolved transition metals and oxides on the surface was investigated but no attempt was made to demonstrate the stability under redox and thermal cycles. In this regard, no obvious advantage over perovskites decorated with transition metals is observed.

5. Moreover, no appropriate durability studies have been made to show the stability of the nanoparticles on the perovskite surface after prolonged operation.

6. Another important aspect is dealing with the use of perovskites anodes for the direct utilisation of hydrocarbons in SOFCs. It would be important to investigate if the synthesised materials can operate in dry methane, as a model organic fuel, and a comparison with relevant literature reports should be made to show if there is any relevant progress beyond the state of the art. In particular, perovskites containing exsolved transition metals have shown already promising capability as an active and stable electrode for SOFCs in various fuels (Nano Lett., 2016, 16 (8), pp 5303–5309) but the comparison should be extended to other perovskites decorated with metal particles or stabilised by thermal treatments for application as anodes in SOFC fed with dry hydrocarbons.

7. In general, the exsolution of transition metals from perovskite backbones in reducing atmosphere has been widely studied in the literature. The authors have mentioned some previous works in this field but relevant literature dealing with anode-based perovskites for SOFCs, including stabilisation procedures and decoration of perovskites with transition metal nanoparticle for application as SOFC anodes is not covered.

Reviewer #3 (Remarks to the Author):

This report shows good performance and nice nanoparticle production. The decomposition of these layered phases is novel although the nature of the exsolution could be better described. The term periodic seems to mean following the periodic table rather than something much more exciting that I was anticipating. This periodicity is not that surprising and could be more or less predicted from standard free energy considerations, perhaps iron is a little out of step. I am not too sure how much the dft adds to this. It would be better if the exsolution was from a single phase rather than from a hexagonal /cubic perovskite mix. Which phase yields the metals. The comparison lacks some more competitive papers, more care could have been taken.

Reviewers' comments:

Reviewer #1 (Remarks to the Author):

General Comments: This manuscript discusses the "periodicity" in the exsolution of transition metal nanoparticle on the layered perovskite oxide and the co-segregation energy. While the first concept is elusive, the second one has been proposed by others. While I believe that the study provides significant new insight on the exsolution process, there are some critical issues need to be considered. I list them here:

Comment 1. At line 77, the authors claim there isn't any secondary phase, however the hexagonal phase is indeed a secondary phase.

Response to C1: The crystal chemistry of these perovskite oxides is in fact complex due to the formation of hexagonal BaMnO₃-type perovskites when synthesis is carried out in air. Raveau *et al.* [*J. Phys.: Condens. Matter*, **14**, 1297 (2002)] and Sengodan *et al.* [*Nat. Mater.*, **14**, 205 (2015)] have reported that the R_xBa_{1-x}MnO₃ (R=La, Pr and Nd) compound is generally a mixture of two main phases namely hexagonal BaMnO₃ and cubic perovskites. This hexagonal perovskite phase increases with increasing the Ba content.

In other words,

Case 1. When $x = 1$, RMnO₃ is pure hexagonal perovskites (Fig. a)

Case 2. When $x = 0.65$, R_{0.65}Ba_{0.35}MnO₃ is mixtures of cubic and hexagonal perovskites (Fig. b).

Case 3. When $x = 0.35$, R_{0.35}Ba_{0.65}MnO₃ is pure cubic perovskite (Fig. c).

Hence, hexagonal perovskite phase is the characteristic phase of Pr_{0.5}Ba_{0.5}MnO₃ and is not a secondary phase (impurity phase).

Comment 2. At line 88, it is easy to understand that MnO, Co, and Ni are more easily exsolved than Fe. However, it is not clear what the authors mean by “periodicity” as described in the following sentence.

Response to C2: Thanks for the comment. In the context of chemistry, periodicity is defined as “trends or recurring variations in element properties”. In our manuscript, the periodicity means trend of exsolution among the B-site doping elements (Mn, Co, Ni and Fe). To avoid the misunderstanding, the text has been modified appropriately to “These results clearly show that MnO, Co, and Ni are more easily exsolved to form nanoparticles than Fe, and thus B-site transition metals show some trend to exsolve in the layered perovskite oxide.”

Comment 3. At line 93 and Figure 2, the SEM image after exsolution shows not only the exsolution of the nanoparticles but also a flatter surface. The small clusters of the host material also vanish after exsolution. This is not in agreement with the authors' comment that the surfaces are similar. Explanations should be provided.

Response to C3: In the SEM image after exsolution, the formation of flatter surface and removal of the small clusters from the host material are due to the phase transition of simple perovskite to layered perovskite. We did not claim that the surface of simple perovskite (before exsolution) and layered perovskite (after exsolution) are similar. In contrast, we claim that the surface morphologies of the reduced samples (L-PBMO, L-PBMCO, L-PBMNO, and L-PBMFO) are similar.

Comment 4. At line 152, the authors assume that the exsolution has two steps, where the metal oxide is formed first. Although this is hard to probe using DFT calculations, I suggest a more careful assumption.

Response to C4: Until now, many mechanisms have been suggested to explain how the B-metal could be exsolved from the parent perovskites. In particular, both formation of A-site deficiency and oxygen vacancy have been suggested to facilitate the exsolution of B-metal on perovskites due to the destabilization of chemical equilibrium or charge state from the broken stoichiometry of perfect perovskite structure [*Nat. Chem.*, **5**, 916 (2013), *Energy Environ. Sci.*, **6**, 256 (2013), *J. Mater. Chem. A*, **3**, 11048 (2015)]. Since all of those papers proposed that oxygen vacancy formation or BO_x formation are essential to the exsolution of dopant metal. Especially, Katz *et al.* suggested from their Pd exsolution study in LaFeO_3 that certain types of defects may be necessary for Pd diffusion, supporting that BO_x complex is closely related to the B-metal movements [*J. Am. Chem. Soc.*, **133**, 18090 (2011)]. Therefore, we could reasonably assume two key sequential steps; BO_x phase segregates toward the surface, and then phase transition from the segregated phase of BO_x to metallic phase might result in the formation of metallic exsolved nanoparticles.

Comment 5. At line 178 and Figure 2, the author seems to only consider the segregation on the (001) plane. This is especially doubtful since PBM is a layered double perovskite and the oxygen transportation may differ quite a lot along different orientations. Calculation for other planes should be provided.

Response to C5: Thanks for your recommendation. Recently, the different degree of B-metal exsolution from parent perovskite structure on different facets has been reported by several papers [*Nat. Comm.*, **6**, 8120 (2015), *Nano Energy*, **27**, 499 (2016)]. Especially, Gao *et al.* performed DFT calculations combined with the experiments of Ni exsolution from $\text{La}_{0.4}\text{Sr}_{0.4}\text{Sc}_{0.9}\text{Ni}_{0.1}\text{O}_{3-\delta}$ (LSSN) catalyst [*Nano Energy*, **27**, 499 (2016)]. They examined Ni segregation energy on Ni (100), (010), (001), and (121) surfaces based on their experimental results of XRD peaks. However, although the degree of segregation energy is different depending on the type of surface, Ni prefers to segregate toward the surface for *all surfaces*. Nonetheless, if necessary, we will further examine the segregation energies on the other promising surfaces for B-metal exsolution based on our XRD data.

Comment 6. The more favorable presence of vacancies at the surface may lead to a coulombic repulsion with other positively charged defects and further promote the BaO formation. This should be explained in greater detail as I believe it is important for the clarification of the mechanism.

Response to C6: Thanks for your consideration. The DFT total energy calculations intrinsically consider the coulombic repulsion between charged defects and oxygen vacancy in the system. Therefore, by comparing the segregation energy of Pr or Ba compared to Co, we can provide more information for the competitive formation of PrO or BaO in addition to that of BO_x . In addition, no BaO formation was detected in experimental results (XRD and TEM).

Comment 7. It would be useful to know which percentage of the B-site dopant migrates to the surface as this information could elucidate if there is an over-stoichiometry of Pr and Ba on the surface itself. Also, why don't BaO or Pr₂O₃ form on the surface as a consequence of exsolution? (this could be easily characterized by XPS).

Response to C7: Thank you for your consideration and recommendation. We measured XPS to know which percentage of the B-site dopant migrates to the surface in L-PBMO, L-PBMC0, L-PBMNO, and L-PBMFO. In case of L-PBMO and L-PBMFO, since Mn and Fe have no metallic phase confirmed by XPS, XRD, and TEM results (Figure b and Figure c), it is hard to calculate the percentage of B-site dopant (Mn and Fe) migration (Bulk and exsolved nanoparticle have oxide phase). And in case of cobalt in L-PBMC0, it is not easy to obtain exact oxidation state of Co because binding energy peaks of Co and Ba overlap each other. Here, we calculated the percentage of B-site dopant (Ni) migration in L-PBMNO. As shown in the Figure a, the percentages of metallic Ni and Ni²⁺ are about 58 and 42 %, respectively, indicating that approximately 58 % of Ni migrates to the surface because

exsolved Ni nanoparticle exists in metallic phase as confirmed by XRD and TEM analyses.

As mentioned in manuscript (Fig.2), after the exsolution of B-site dopants from the host perovskite lattice, excess BaO and Pr₂O₃ are not formed due to the formation of B-site vacant layered perovskite (Schottky type defect) as follows:

Where Pr_{Pr}^{\times} indicates Pr ions in Pr site with net charge zero, Ba_{Ba}^{\times} indicates Ba ions in Ba site with net charge zero, Mn_{Mn}^{\times} indicates Mn ions in Mn site with net charge zero, T_{Mn}^{\times} indicates T (Co, Fe, Ni and Mn) ions in Mn site with net charge zero, O_O^{\times} indicates O ions in O site with net charge zero, V_{Mn}'' indicates metal ions vacancy in Mn site with net charge -2, and $V_O^{\circ\circ}$ indicates oxygen ions vacancy in O site with net charge +2.

Reviewer #2 (Remarks to the Author):

General Comments: This paper deals with the exsolution of transition metals in layered perovskites. The novel aspect should regard the so called “peridocioty of exsolution” in redox stable layered perovskites i.e. a correlation between the thermodynamic reducibility of transition metal cations used as B-site dopants and the occurrence of the corresponding metal and oxide nanoparticles on the surface. My comments are provided below for the authors’ consideration.

Comment 1. The relationship between the thermodynamic reducibility of transition metals cations used as B-site dopants and the occurrence of the corresponding metal and oxide nanoparticles on the surface is not unexpected. At high temperature, it is obvious that solid state reactions are essentially governed by thermodynamics. However, the authors have carried out a systematic work providing a wide demonstration of what is generally expected and experimentally verified in specific cases in other reports. Anyhow, the general analysis here provided could be of interest for SOFC developers.

Response to C1: We thank reviewer #2 for generally supporting our paper.

Comment 2. Whereas the effect of oxygen vacancies and A-site deficiency has been previously investigated [*Nat. Chem.*, **5**, 916 (2013)] and the new contribution provided in this regard by this paper, using DFT calculations, should be better elucidated in comparison to the previous literature reports. Moreover, in a previous work recently published, density functional theory calculations have also proved that the crystal reconstruction induces the loss of coordinated oxygen surrounding B-site cations, serving as the driving force for steering fast nanoparticle growth (*Nano Lett.*, 2016, 16 (8), pp 5303–5309). This previous work is not mentioned in the manuscript.

Response to C2: We thank for the reviewer's suggestion. Although our DFT calculations are well explained in the comparisons with the previous literature reports (ref. 23), it is hard to compare our stoichiometric layered perovskite system directly with the A-site deficient simple perovskite system [*Nat. Chem.*, **5**, 916 (2013)]. In terms of exsolution phenomenon, the literature [*Nano Lett.*, **16**, 5303 (2016)] has been cited appropriately.

Comment 3. The occurrence of metal or oxide nanoparticle as consequence of the exsolution is essentially demonstrated in this work by using X-ray diffraction and transmission electron microscopy with EDS analysis. TEM analysis provides a local information involving just a few nanoparticles. However, there is no proper analysis of the surface modifications on a statistical basis after the exsolution in terms of composition and oxidation states. On the other hand, a quantitative determination of the surface atomic composition and oxidation states may be provided by X-ray photoelectron spectroscopy analysis. The catalytic activity is essentially related the surface characteristics since reactions occur at the interface; thus, I strongly recommend a surface analysis e.g. by XPS for this work.

Response to C3: Thank you for your consideration. According to your recommendation, we measured XPS to obtain the information of surficial chemical composition in L-PBMO, L-PBMCO, L-PBMNO, and L-PBMFO. As shown in the Figure a, the binding energy peaks of metallic Ni (855.3 and 873.1 eV) could be detected in L-PBMNO, which is consistent with the XRD and TEM results. Figure b presents the binding energy peaks of Mn^{2+} (641 and 652.8 eV) and Mn^{3+} (642.8 and 653.7 eV) in L-PBMO. Figure c presents two major peaks with binding energy at 710.2 and 723.9 eV, corresponding to Fe $2p_{3/2}$ and Fe $2p_{1/2}$, accompanied by two shake-up satellite peaks (718.6 and 732 eV) in L-PBMFO. The appearance of two peaks at 710 and 723.7 eV is Fe^{2+} , whereas the other two peaks at 712.5 and 725.5 eV are characteristic of Fe^{3+} . From the XPS results, we can know that Mn and Fe have no metallic phase in L-PBMO and L-PBMFO, respectively, which is also in agreement with our experimental results. However, in case of cobalt in L-PBMCO (Figure d), it is not easy to obtain exact oxidation state of Co because binding energy peaks of Co and Ba overlap each other. [*Adv. Mater.*, **28**, 6442 (2016), *J. Memb. Sci.*, **382**, 158 (2011)].

Comment 4. The aim of using catalysts made of perovskite with exsolved transition metal on the surface in SOFCs is to provide an enhanced activity, being this promoted by the metal nanoparticles on the surface, while keeping good stability towards redox and thermal cycles. The performance of hydrogen-fed SOFCs for the perovskites containing exsolved transition metals and oxides on the surface was investigated but no attempt was made to demonstrate the stability under redox and thermal cycles. In this regard, no obvious advantage over perovskites decorated with transition metals is observed.

Response to C4: Thanks for the comments. The layered perovskite anodes have been developed by various groups and all reports shows excellent redox and thermal cycles [*Nat. Mater.*, **14**, 205 (2015), *Nano Lett.*, **16**, 5303 (2016), *J. Power Sources.*, **342**, 313 (2017)]. In addition, our work was not for performance (redox and thermal cycles) of SOFCs in different fuels but for understanding the exsolution phenomenon of different catalytically active transition metals in layered perovskite anodes. Few points should be emphasized in terms of novelty and uniqueness of this work. This is the first work to show competitive exsolution phenomenon of different transition metals in stoichiometric layered perovskite anodes. The second point is that co-segregation of B-site metal dopant and oxygen vacancies plays central role in the exsolution.

Comment 5. Moreover, no appropriate durability studies have been made to show the stability of the nanoparticles on the perovskite surface after prolonged operation.

Response to C5: In this work, we focused on phenomenon in terms of periodicity of exsolution. In addition, $\text{PrBaMn}_{1.7}\text{T}_{0.3}\text{O}_{5+\delta}$ (T= Co and Ni) perovskite anodes decorated with metal nanoparticles show high performance in H_2 fuel compared to other ceramic anodes. However, we will include the stability test in H_2 and C_3H_8 at 700°C for further consideration.

Comment 6. Another important aspect is dealing with the use of perovskites anodes for the direct utilisation of hydrocarbons in SOFCs. It would be important to investigate if the synthesised materials can operate in dry methane, as a model organic fuel, and a comparison with relevant literature reports should be made to show if there is any relevant progress beyond the state of the art. In particular, perovskites containing exsolved transition metals have shown already promising capability as an active and stable electrode for SOFCs in various fuels (Nano Lett., 2016, 16 (8), pp 5303–5309) but the comparison should be extended to other perovskites decorated with metal particles or stabilised by thermal treatments for application as anodes in SOFC fed with dry hydrocarbons.

Response to C6: In this work, we focused phenomenon in terms of periodicity of exsolution. In addition, $\text{PrBaMn}_{1.7}\text{T}_{0.3}\text{O}_{5+\delta}$ (T= Co and Ni) perovskite anodes decorated with metal nanoparticles show high performance in H_2 fuel compared to other ceramic anodes. However, we will include the electrochemical performance in C_3H_8 at $800\text{ }^\circ\text{C}$ for further consideration.

Comment 7. In general, the exsolution of transition metals from perovskite backbones in reducing atmosphere has been widely studied in the literature. The authors have mentioned some previous works in this field but relevant literature dealing with anode-based perovskites for SOFCs, including stabilisation procedures and decoration of perovskites with transition metal nanoparticle for application as SOFC anodes is not covered.

Response to C7: We have covered stabilization procedure to stabilize the simple perovskite structure in the introduction section (ref. 12). For example, “the control of non-stoichiometry (A-site deficiency in the ABO_3 stoichiometry) to promote exsolution”.

Reviewer #3 (Remarks to the Author):

General Comments: This report shows good performance and nice nanoparticle production. The decomposition of these layered phases is novel although the nature of the exsolution could be better described. The term periodic seems to mean following the periodic table rather than something much more exciting that I was anticipating. This periodicity is not that surprising and could be more or less predicted from standard free energy considerations, perhaps iron is a little out of step. I am not too sure how much the dft adds to this. It would be better if the exsolution was from a single phase rather than from a hexagonal /cubic perovskite mix. Which phase yields the metals. The comparison lacks some more competitive papers, more care could have been taken.

Response to Comments: Thanks for your consideration. Our DFT results can play an important role in decoupling the mechanism of exsolution phenomenon at the atomic scale. Under the reasonable hypothesis, we divided the exsolution behavior into the two key elementary steps using surface models; 1) segregation of BO_x phase toward the surface and 2) phase transition from the segregated phase of BO_x to metallic phase. Our DFT results are in good agreement with the experiment results, indicating that our hypothesis seems plausible and the two key elementary steps may dominate the exsolution behavior even if there exist other factors for the exsolution.

In the manuscript, we clearly discussed that exsolution of transition metal occurs only in layered perovskite $\text{PrBaMn}_{1.7}\text{T}_{0.3}\text{O}_{5+\delta}$ (T= Mn, Co, Ni, and Fe) rather than hexagonal /cubic perovskite.

Reviewers' comments:

Reviewer #1 (Remarks to the Author):

I am satisfied with the authors replies and addressing all questions in an exhaustive, clear and concise manner. I am happy to see that the article was improved. I fully support the publication of this article in present form

Reviewer #2 (Remarks to the Author):

The authors have properly addressed most of my previous observations. However, some points have not been carefully considered.

Although, the focus of the paper is on periodicity of exsolution, the endurance tests in hydrogen and organic fuel represent an important validation of the stability of the materials under operation. Durability tests of half day or one day, such as those presented in the reply to reviews, are not sufficient to demonstrate proper stability for these materials. This time span just corresponds to the conditioning period of a SOFC, whereas a reliable endurance test should not be less than 150 hrs or a week both in H₂ and organic fuels.

Comparison with other perovskites decorated with metal particles or stabilised by thermal treatments for application as anodes in SOFC fed with dry hydrocarbons has not been addressed.

Reviewers' comments:

Reviewer #1 (Remarks to the Author):

General Comments: I am satisfied with the authors replies and addressing all questions in an exhaustive, clear and concise manner. I am happy to see that the article was improved. I fully support the publication of this article in present form.

Response to Comments: We sincerely appreciate the reviewer for evaluating our manuscript and providing valuable comments and suggestions that benefit our manuscript.

Reviewer #2 (Remarks to the Author):

General Comments: The authors have properly addressed most of my previous observations. However, some points have not been carefully considered. Although, the focus of the paper is on periodicity of exsolution, the endurance tests in hydrogen and organic fuel represent an important validation of the stability of the materials under operation. Durability tests of half day or one day, such as those presented in the reply to reviews, are not sufficient to demonstrate proper stability for these materials. This time span just corresponds to the conditioning period of a SOFC, whereas a reliable endurance test should not be less than 150 hrs or a week both in H₂ and organic fuels. Comparison with other perovskites decorated with metal particles or stabilized by thermal treatments for application as anodes in SOFC fed with dry hydrocarbons has not been addressed.

Response to Comments: Thank you for your consideration. We include electrochemical performance and stability test in H₂ and C₃H₈. No degradation was observed under a constant voltage of 0.6 V at 700 °C in H₂ and C₃H₈ for more than 200 h. Furthermore, we compare our work with other perovskites decorated with metal particles or stabilized by thermal treatments for application as anodes with hydrocarbons. A comparison of fuel cell performance of other perovskites decorated with metal particles anodes in hydrocarbon fuel has been included in the manuscript (Supplementary Table 3).

Figure. (a) I-V curve and the corresponding power densities of the L-PBMCO and L-PBMNO electrode using C₃H₈ as fuel and ambient air as the oxidant at 800 °C. Electrochemical performances of L-PBMNO anode in (b) H₂ and (c) C₃H₈ at 700 °C under a constant voltage of 0.6 V.

REVIEWERS' COMMENTS:

Reviewer #2 (Remarks to the Author):

I am fine with the replies from the authors. However, since the durability tests in hydrocarbons represent an important result of this work, I strongly recommend to move both Fig. 11 a-c, presently reported in Supplementary Information, and the comparison with the literature, from Supplementary Information to the main text to clearly show the progress beyond the state of the art.

Reviewers' comments:

Reviewer #2 (Remarks to the Author):

General Comments: I am fine with the replies from the authors. However, since the durability tests in hydrocarbons represent an important result of this work, I strongly recommend to move both Fig. 11 a-c, presently reported in Supplementary Information, and the comparison with the literature, from Supplementary Information to the main text to clearly show the progress beyond the state of the art.

Response to Comments: We sincerely appreciate the reviewer for evaluating our manuscript and providing valuable comments and suggestions that benefit our manuscript. As recommended by the reviewer, we moved the supplementary figure 11 (electrochemical performance and durability of fuel cells in hydrocarbon fuels) to the main text. Due to space restrictions, we decided to leave the supplementary table 3 (comparison with the literature) in supplementary file.